# Phenelzine and Amoxapine Inhibit Tyramine and d-Glucuronic Acid Catabolism in Clinically Significant *Salmonella* in A Serotype-Independent Manner

**DOI:** 10.3390/pathogens10040469

**Published:** 2021-04-13

**Authors:** Raquel Burin, Devendra H. Shah

**Affiliations:** 1Department of Veterinary Microbiology and Pathology, College of Veterinary Medicine, Washington State University, Pullman, WA 99164, USA; raquel.burin@wsu.edu; 2Paul Allen School for Global Animal Health, College of Veterinary Medicine, Washington State University, Pullman, WA 99164, USA

**Keywords:** *Salmonella*, d-glucuronic acid, tyramine, phenelzine, amoxapine

## Abstract

Non-typhoidal *Salmonella* ingeniously scavenges energy for growth from tyramine (TYR) and d-glucuronic acid (DGA), both of which occur in the host as the metabolic byproducts of the gut microbial metabolism. A critical first step in energy scavenging from TYR and DGA in *Salmonella* involves TYR-oxidation via TYR-oxidoreductase and production of free-DGA via β-glucuronidase (GUS)-mediated hydrolysis of d-glucuronides (conjugated form of DGA), respectively. Here, we report that *Salmonella* utilizes TYR and DGA as sole sources of energy in a serotype-independent manner. Using colorimetric and radiometric approaches, we report that genes *SEN2971*, *SEN3065*, and *SEN2426* encode TYR-oxidoreductases. Some *Salmonella* serotypes produce GUS, thus can also scavenge energy from d-glucuronides. We repurposed phenelzine (monoaminoxidase-inhibitor) and amoxapine (GUS-inhibitor) to inhibit the TYR-oxidoreductases and GUS encoded by *Salmonella*, respectively. We show that phenelzine significantly inhibits the growth of *Salmonella* by inhibiting TYR-oxidoreductases SEN2971, SEN3065, and SEN2426. Similarly, amoxapine significantly inhibits the growth of *Salmonella* by inhibiting GUS-mediated hydrolysis of d-glucuronides. Because TYR and DGA serve as potential energy sources for *Salmonella* growth in vivo, the data and the novel approaches used here provides a better understanding of the role of TYR and DGA in *Salmonella* pathogenesis and nutritional virulence.

## 1. Introduction

Host–microbiota interactions within the gastrointestinal (GI) tract have profound effects on the physiology and immune system of the host. Turbulence in the commensal microbiota of the GI tract plays an important role in aiding the survival, growth, and persistence of pathogens, such as *Salmonella,* within this niche. Several factors including the production of inhibitory substances of microbial origin, competition for adhesion sites, and acquisition of nutrients contribute to protective mechanisms afforded by the GI microbiota. For the efficient acquisition of nutrients within the nutritionally challenging environment of the GI tract, *Salmonella* has evolved to scavenge energy from the metabolic byproducts of the commensal microbiota. The metabolic versatility of *Salmonella* is therefore increasingly recognized as a pathogenicity factor [1,2,3,4,5,6,7,8,9,10]. However, the precise metabolic requirements of *Salmonella* in the host GI tract remain poorly defined. Therefore, identification of metabolic byproducts and the knowledge of the metabolic pathways used by *Salmonella* to scavenge energy from such byproducts is essential to better understand the mechanisms underlying the nutritional virulence of *Salmonella.*

Recently, we reported that *Salmonella* can ingeniously scavenge energy from tyramine (TYR, an aromatic amine) and d-glucuronic acid (DGA, a hexuronic acid), both of which are known to naturally occur in the host as byproducts of gut microbial metabolism [11]. Within the mammalian GI tract, TYR is produced as a result of tyrosine (a semi-essential amino acid) decarboxylation, a reaction mediated by the gut microbial enzyme tyrosine decarboxylase [12,13,14,15,16]. TYR is also naturally found in fermented food products [12]. As a result, varying concentrations of TYR can be detected in cecal contents (0.3 to 0.9 µM) and other host tissues (1.5 to 32 ng/g) [17,18]. Interestingly, enteric bacteria such as *E. coli* and *Salmonella* can utilize TYR as a source of energy for growth [11,19,20,21,22,23]. In *E. coli,* TYR serves as a substrate for oxidoreductase enzyme TynA (also known as monoamine oxidoreductase or MAO). TynA oxidizes TYR in a reaction that liberates hydrogen peroxide and ammonia, thereby activating the TYR catabolic pathway which shunts the resulting pyruvate and acetyl-CoA into the TCA cycle [20,22]. TynA mediated TYR oxidation is considered a critical first step in the TYR catabolic pathway. It has been reported that the *tynA* deletion mutant of *E. coli* fails to grow when TYR is used as a sole source of energy [21]. We reported that *Salmonella* can also derive energy from TYR as a sole source of carbon and nitrogen; however, *tynA* gene is not present in this pathogen [11,23]. Thus, to identify genes encoding alternative TYR oxidoreductases, we recently conducted TYR-induced transcriptome profiling of *Salmonella* [23]. In that study, we identified three TYR-responsive genes (*SEN2971, SEN3065,* and *SEN2426*) (Appendix A) and reported that these genes encode putative oxidoreductase enzymes [23]. We also reported that deletion of *SEN2971* significantly impairs, but does not eliminate the ability of *Salmonella* to efficiently derive energy for growth from TYR, suggesting that SEN2971 likely serves as a primary TYR oxidoreductase [11,23], whereas proteins encoded by *SEN3065* and *SEN2426* likely serve as alternative TYR oxidoreductases [23]. However, it is currently unknown if *SEN2971, SEN3065,* and *SEN2426* are indeed TYR-oxidoreductases. Because oxidoreductase-mediated TYR oxidation is the critical first step in the TYR catabolism, here we aimed to determine whether non-typhoidal *Salmonella* (NTS) utilize TYR as a sole source of energy for growth, in a serotype-independent manner and whether SEN2971, SEN3065, and SEN2426 are TYR oxidoreductases. Additionally, we aimed to test whether inhibition of oxidoreductase activity of these newly identified TYR oxidoreductases could be used as a strategy to block the ability of *Salmonella* to derive energy from TYR in a serotype-independent manner. To test this hypothesis, we repurposed phenelzine, a broad-spectrum MAO inhibitor (MAOi) known to inhibit MAO-mediated oxidation of other monoamine neurotransmitters such as dopamine, serotonin, and norepinephrine [24,25]. 

Unlike TYR, DGA is a carboxylic acid derived from the oxidation of glucose, and it commonly occurs in a conjugated form (known as d-glucuronide) in the liver and the mucus layers of the mammalian GI tract [26]. In the liver, conjugation of DGA with drugs and other toxic compounds of endogenous and exogenous origins occurs as the most abundant phase-II reaction (i.e., glucuronidation), and through enterohepatic circulation, these conjugates reach the mammalian GI tract. Glucuronidation increases the water solubility of toxic compounds, makes them less biologically active (inert), and promotes their removal from the body via kidneys or GI tract [27]. Thus, a high concentration of d-glucuronides can be detected in the small intestine (40 to 104 µg/g), cecum (14 to 28 µg/g), and colon (5 to 14 µg/g) [26]. Interestingly, d-glucuronides also serve as a substrate for microbial β-glucuronidase (GUS) enzymes in the GI tract. More specifically, d-glucuronides are hydrolyzed by the GUS enzymes produced by commensal gut microbiota which results in recirculation of the toxic compounds in the enterohepatic cycle and the release of the free-DGA moiety [23,27]. The free-DGA then enters the Entner–Doudoroff pathway (a bacterial alternative to glycolysis) that catabolizes sugar acids and shunts the resulting pyruvate into the TCA cycle. Pathogenic strains of *E. coli* such as enterohemorrhagic *E. coli* and several clinically significant *Salmonella* serotypes including *Salmonella enterica sub sp enterica* serovar Enteritidis (*Salmonella* Enteritidis) and *Salmonella* Typhimurium do not produce GUS enzymes; thus, these pathogens cannot directly hydrolyze d-glucuronide. However, they can utilize free-DGA as an energy source for growth [11,28,29]. In contrast, several commensal or pathogenic strains of *E. coli* and a few serotypes of *Salmonella* such as *S.* Montevideo and *S.* Schwarzengrund carry GUS operon, thus are capable of hydrolyzing d-glucuronide [11,28,29,30]. Therefore, the free-DGA released due to the action of either the microbiota or the pathogen produced GUS serves as an energy source for the propagation of enteric pathogens such as enterohaemorrhagic *E. coli* O157:H7 as well as for *Salmonella* [2,11,23,31]. In summary, GUS enzymes mediate the hydrolysis of d-glucuronide resulting in the reactivation of toxic molecules important in host health and disease, with the simultaneous release of free-DGA, which is then utilized by enteric pathogens as an energy source. Interestingly, in human medicine, there is increased interest in the development and use of GUS inhibitory compounds as a clinical strategy to reduce the hydrolytic breakdown of d-glucuronides by commensal microbiota to reduce reactivation of toxic metabolites in the GI tract [31,32,33,34]. Given that the GUS enzymes are at the interface of a metabolic symbiosis between enteric pathogens and the host microbiota, in this study, we aimed to repurpose amoxapine (a tricyclic GUS-inhibitor and antidepressant of the dibenzoxazepine class) to inhibit the GUS-mediated hydrolysis of d-glucuronides by *Salmonella,* thereby limiting the ability of *Salmonella* to derive energy from d-glucuronides for growth. Here, we show that *Salmonella* ingeniously scavenges energy for growth from TYR and DGA, in a serotype-independent manner. We also report that MAO inhibitor phenelzine and GUS inhibitor amoxapine significantly impair the ability of *Salmonella* to derive energy from TYR and d-glucuronides in vitro and consequently cause a growth defect in *Salmonella* in a serotype-independent manner. Given that TYR and DGA may serve as an important source of energy for *Salmonella* growth in vivo, the data and the novel approaches used in this study will open new avenues to investigate the role of TYR and DGA in *Salmonella* pathogenesis and nutritional virulence in vivo. 

## 2. Results and Discussion

### 2.1. SEN2971, SEN3065, and SEN2426 Are TYR Oxidoreductases

We recently reported that the expression of *SEN2971*, *SEN3065*, and *SEN2426* encoding putative oxidoreductases was highly upregulated when TYR is used as a sole energy source for the growth of *Salmonella* [23]. Previously, it was reported that the expression of *STM3128* (gene corresponding to SEN2971) is also upregulated in response to TYR in *S.* Typhimurium [35]. Moreover, the deletion of *SEN2971* in *S.* Enteritidis and *STM3128* in *S.* Typhimurium was also reported to significantly impair the ability of these two serotypes to utilize TYR as a sole energy source for growth [11,23,35,36,37]. In this study, we produced recombinant SEN2971 (50.87 KDa), SEN3065 (39.93 KDa), and SEN2426 (30.87 KDa) and determined their oxidoreductase activity through a radiometric assay using tritium (^3^H) labeled TYR as a substrate. The results of the radiometric assays show that the oxidoreductase activity of SEN2971 was highest (25 CPM), followed by SEN3065 (10 CPM) and SEN2426 (6 CPM) (Figure 1a). We confirmed the oxidoreductase activity of SEN2971, SEN3065, and SEN2426 using a colorimetric assay that detects the oxidoreductase-dependent formation of H_2_O_2_. As expected, SEN2971 showed the strongest TYR oxidoreductase activity (OD_562_: 0.087) (Figure 1b). The oxidoreductase activity of SEN2971 was similar to H_2_O_2_ as a positive control (OD_562_ = 0.086); however, the oxidoreductase activity of SEN3065 (OD_562_ = 0.015) and SEN2426 (OD_562_ = 0.011) was significantly (*p* < 0.05) lower than SEN2971 (Figure 1b). These results corroborate with the TYR oxidoreductase activity of these recombinant proteins observed with the radiometric assay. Although three recombinant oxidoreductases tested in this study showed differential TYR oxidoreductase activities, it is possible that these differences are likely confounded by potential differences in the recombinant protein concentrations used in the assays. Follow-up studies are needed to confirm the differential efficiency of individual TYR oxidoreductases in *Salmonella*. Nevertheless, the data clearly show that *Salmonella* is capable to efficiently oxidize TYR through the combined actions of three oxidoreductases SEN2971, SEN3065, and SEN2426, and that the oxidoreductase encoded by SEN2971 likely exhibits the highest oxidoreductase activity.

To determine whether SEN2971, SEN3065, and SEN2426 are MAO-A or MAO-B type, we tested differential inhibition of these recombinant oxidoreductases with clorgyline, an inhibitor of the MAO-A activity, and pargyline, as an inhibitor of MAO-B activity, respectively. The oxidoreductase activity of SEN3065 (17%) and SEN2426 (13%) was significantly (*p* < 0.05) lower than SEN2971 (101%) and the positive control (100%) (Figure 2). The TYR oxidoreductase activity of SEN2971 was significantly (*p* < 0.05) reduced by the inhibitor clorgyline (23%), followed by pargyline (64%), suggesting that SEN2971 is likely an MAO-A type (Figure 2). In corroboration with these results, others have also reported that the activity of *E. coli* oxidoreductase TynA is more efficiently inhibited with clorgyline when compared with pargyline, suggesting that TynA is likely MAO-A type [38]. The oxidoreductase activity of SEN3065 (17%) was similar when treated with clorgyline (17%); however, reduced activity was detected when treated with pargyline (13%). These differences were not statistically significant (*p* > 0.05). Similarly, the oxidoreductase activity of SEN2426 (13%) was similar when treated with clorgyline (15%); however, reduced activity was detected when treated with pargyline (8%). These differences were also not statistically significant (*p* > 0.05).

### 2.2. Phenelzine Inhibits the TYR Oxidoreductase Activity of Recombinant SEN2971, SEN3065, and SEN2426

Phenelzine is a broad-spectrum monoamine oxidase inhibitor (MAOi) that inhibits both MAO-A and MAO-B [39]. Phenelzine is also known to inhibit the activity of an *E. coli* MAO, TynA [38]. In the current study, treatment with phenelzine significantly (*p* < 0.05) inhibited the TYR oxidoreductase activity from 25 CPM (SEN2971), 10 CPM (SEN3065), and 6 CPM (SEN2426) down to 4 CPM (SEN2971 and SEN3065) and <1 CPM (SEN2426), respectively (Figure 3). These data show that phenelzine can efficiently inhibit the oxidoreductase activity of all three TYR oxidoreductases SEN2971, SEN3065, and SEN2426. Consequently, it can be expected that the use of phenelzine may abrogate the ability of *Salmonella* to scavenge energy from TYR and may significantly impair the growth of *Salmonella* when TYR is supplemented as the sole energy source.

### 2.3. Phenelzine Inhibits the Growth of S. Enteritidis in the Presence of TYR as the Sole Energy Source

Previously, we reported that deletion of *SEN2971* significantly impaired the growth, but did not abrogate the ability of *Salmonella* to grow by deriving energy from TYR [11]. We hypothesized that the residual growth of *S.* Enteritidis Δ*SEN2971* was likely due to the activity of alternative oxidoreductases encoded by TYR-inducible genes *SEN3065* and *SEN2426* [23]. Because phenelzine is a broad-spectrum MAO inhibitor that inhibits the activity of SEN2971, SEN3065, and SEN2426 (Figure 3), we reasoned that treatment with phenelzine should inhibit any residual growth of Δ*SEN2971* due to the actions of alternative oxidoreductases SEN3065 and SEN2426. To test this hypothesis, we tested the growth of WT *S.* Enteritidis str. 1 and Δ*SEN2971* in the medium supplemented with TYR as a sole energy source, with or without the treatment with phenelzine. The untreated WT of *S.* Enteritidis grew up to 4.16 log_10_ during 48 h with a doubling time (Td) of 3.52 h (Figure 4a). As expected, the growth of untreated *S.* Enteritidis Δ*SEN2971* was significantly (*p* < 0.05) lower (1.89 log_10_) with a Td of 7.62 h when compared with the untreated WT strain. The residual growth of Δ*SEN2971* here can be attributed to the activity of two alternative oxidoreductases (SEN3065 and SEN2426). When WT and Δ*SEN2971* were treated with 30 µM of phenelzine, the Td of phenelzine treated WT (18.80 h) increased significantly when compared with non-treated WT (3.52 h). Similarly, the Td of Δ*SEN2971* increased significantly from 7.62 h to 29.49 h (Figure 4a). These results clearly show that the inhibition of the residual growth of Δ*SEN2971* is likely due to the inhibition of the oxidoreductase activity of SEN2426 and SEN3065. The inhibition of *S.* Enteritidis growth is unlikely due to the off-target effects of phenelzine because WT and Δ*SEN2971* treated with 30 µM of phenelzine grew up to 6 log_10_ with a Td of 2.26 h and 2.25 h, respectively, when TYR was replaced with glucose as a sole source of energy (Figure 4b). These data show that phenelzine is not toxic to the bacterial cells and that this inhibitor can specifically block the utilization of TYR as a source of energy by *Salmonella* by inhibiting TYR-oxidoreductases.

### 2.4. Clinically Significant NTS Utilize TYR as a Source of Energy

Previously we reported that *S.* Enteritidis can utilize TYR as a sole energy source for growth [11,23]. To determine whether TYR is also utilized by other NTS serotypes as an energy source, we tested the ability of twelve *Salmonella* serotypes to utilize TYR as a source of energy for growth, as previously determined for the WT *S.* Enteritidis str. 1 as a model organism. These 12 serotypes were recently reported as the most-prevalent poultry-associated *Salmonella* serotypes (MPPSTs) and all, except *S.* Kentucky were also identified as the clinically most significant serotypes associated with food-borne illnesses in humans in the USA [40,41]. All the *Salmonella* strains showed growth in M9 containing 5 mM of TYR (without NH_4_Cl) ranging from 1.42 (*S.* Thompson) to 5.08 (*S.* I 4, 5, 12: I:-) log_10_ within 48 h (Figure 5). A few serotypes such as *S.* Thompson (1.42 log_10_) and *S.* Mbandaka (1.52 log_10_) showed lower growth when compared with *S.* Enteritidis str. 1, whereas other serotypes such as *S.* Seftenberg (4.98 log_10_), *S.* Schwarzengrund (5.06 log_10_), and *S.* I4,5,12:I:- (5.08 log_10_) showed higher growth than *S.* Enteritidis str. 1. These data suggest that all serovars utilize TYR as energy source for growth with differential efficiency. We tested each serovar for the presence of *SEN2971*, *SEN3065*, and *SEN2426* genes encoding oxidoreductase enzymes. All serovars showed positive amplification of each gene, suggesting that these genes are present within the genome of each serovar (Appendix A). It is currently unknown if the differential growth in TYR supplemented media is due to the differential expression of these genes or the differential ability of these serovars to degrade acetaldehyde, in which the rate of acetaldehyde synthesis may exceed the rate of acetaldehyde consumption leading to toxic effects in some serotypes [23]. Despite some differences in the growth rates, these results show that all serotypes tested in this study can grow by utilizing TYR as a sole energy source.

### 2.5. Phenelzine Inhibits the Ability of NTS to Utilize TYR as a Sole Energy Source in a Serotype-Independent Manner

Given that all *Salmonella* serotypes tested in this study utilize TYR as a sole source of energy for growth and that all serotypes carry *SEN2971*, *SEN3065*, and *SEN2426* genes encoding oxidoreductase enzymes, we tested whether phenelzine treatment inhibits the growth of these *Salmonella* strains when TYR is supplemented as a sole energy source. In this assay, the untreated *Salmonella* strains grew with a Td of 9.48 h (*S.* Thompson) to 4.72 h (*S.* I 4, 5, 12:I:-) (Table 1). The treatment with 30 µM of phenelzine significantly increased the Td ranging from 13.89 h (*S.* Thomson) to 14.55 h (*S.* I 4, 5, 12:I:-), respectively (Table 1). These results show that phenelzine significantly inhibits the utilization of TYR as an energy source in NTS in a serotype-independent manner.

### 2.6. Clinically Significant NTS Utilize Free-DGA as a Sole Source of Energy for Growth in a Serotype-Independent Manner

Previously, we reported that *S.* Enteritidis can utilize free-DGA as a sole source of energy for growth [11,23]. In this study, we tested the growth of 11 additional NTS serotypes in M9 containing 1 mM of DGA as a sole source of energy for growth. In this assay, all the *Salmonella* strains showed growth ranging from 6.79 log_10_ (*S.* Seftenberg) to 9.38 log_10_ (*S.* Typhimurium) within 24 h (Figure 6). These data suggest that all NTS serotypes utilize free-DGA as a sole source of energy for growth in a serotype-independent manner.

### 2.7. NTS Hydrolyze d-Glucuronide in a Serotype-Dependent Manner

We previously reported that not all NTS serotypes carry GUS operon, thus *Salmonella* serotypes that carry GUS operon may hydrolyze d-glucuronide resulting in the production of free-DGA which is then utilized as an energy source for growth [11]. To determine the differential ability of NTS serotypes to hydrolyze d-glucuronide, we first performed a cell-based colorimetric assay that measures the level of dissociation of the d-glucuronide PNPG (colorless) to p-nitrophenol (yellow). In this assay, we used *S.* Enteritidis str. 1, which lacks GUS operon, and *S.* Montevideo str. 2, which carries GUS operon [11,42] as reference test strains (Table 2). 

As expected, p-nitrophenol production was not detected in a reaction containing *S.* Enteritidis, suggesting that this serotype does not produce GUS and thus cannot hydrolyze d-glucuronide PNPG (Figure 7a). In contrast, a significantly (*p* < 0.05) high amount of p-nitrophenol was detected in a reaction containing *S.* Montevideo, suggesting that *S.* Montevideo produces GUS and therefore this serotype can efficiently hydrolyze d-glucuronide PNPG (Figure 7a). Next, we tested the ability of the *S.* Enteritidis str. 1 and *S.* Montevideo str. 2 to hydrolyze p-Acetamidophenyl β-d-glucuronide sodium salt (d-glucuronide AA-Gluc) as an energy source for growth. d-glucuronide AA-Gluc was chosen for the growth assay because the hydrolysis of d-glucuronide PNPG used in the cell-based colorimetric assay releases p-nitrophenol, a compound that inhibits bacterial growth (data not shown). For the growth assay, *S.* Enteritidis str. 1 and *S.* Montevideo str. 2 were cultured for 24 h in M9 containing 1 mM of d-glucuronide AA-Gluc as a sole energy source. As expected, the growth of *S.* Montevideo was significantly (*p* < 0.05) higher (4.82 log_10_ with a Td of 2.41 h) when compared with *S.* Enteritidis (0.56 log_10_ with a doubling time of 5.98 h) (Figure 7b), indicating that the lack of *S.* Enteritidis growth in this assay is due to the lack of GUS. We then tested the growth of twelve clinically significant NTS serotypes in M9 containing 1 mM of d-glucuronide AA-Gluc as a sole energy source. Two serotypes, *S.* Schwarzengrund (3.93 log_10_) and *S.* Montevideo (5.07 log_10_) grew at levels similar to that of the GUS producing *S.* Montevideo str. 2 (5.28 log_10_) (Figure 7c). However, all other NTS serotypes showed significant growth defects, suggesting that due to the lack of GUS these serotypes cannot scavenge free-DGA via hydrolysis of d-glucuronide AA-Gluc. Collectively, these data show that NTS differ in their ability to hydrolyze d-glucuronide, and, as a result, show a differential ability to scavenge energy from d-glucuronide for growth.

### 2.8. Amoxapine Inhibits the GUS-Mediated Hydrolysis of d-Glucuronide PNPG by S. Montevideo

Amoxapine is a tricyclic antidepressant of the dibenzoxazepine class which has been reported to exhibit broad-spectrum inhibitory activity against bacterial GUS including *E. coli* [31,34]. Therefore, we tested whether GUS-mediated hydrolysis of d-glucuronide PNPG by *S.* Montevideo can also be inhibited by amoxapine. In the case of *S.* Montevideo, the GUS mediated hydrolysis of d-glucuronide PNPG was inhibited by amoxapine in a concentration-dependent manner with the highest inhibitory activity detected at concentrations of ≥10 µM (Figure 8a). A GUS-negative *S.* Enteritidis str. 1 was used as a control in this assay. As expected, the GUS activity was not detected in *S.* Enteritidis at any concentration of amoxapine used (Figure 8a). A purified β-glucuronidase from *E. coli* was also used as a comparison in the colorimetric assay. *E. coli* GUS revealed a strong activity (OD_450_ = 0.637) which was completely inhibited by 50 µM of amoxapine (Figure 8b). Cell survivability was also assessed by plating 10-fold dilutions of the cultures treated with varying concentrations of amoxapine (1 µM to 100 µM). At 10 µM of amoxapine, the survivability of *S.* Enteritidis and *S.* Montevideo was 100% and 94%, respectively (Figure 8c). To rule out the off-target effects of amoxapine at 10 µM concentration, we tested the growth of *S.* Enteritidis and *S.* Montevideo in the presence of free-DGA as a sole source of energy with or without the addition of 10 µM of amoxapine. There was no significant difference in the overall growth or the Td of either serotype in the presence or absence of amoxapine when free-DGA was supplemented as an energy source (Figure 8d). Collectively, these data show that 10µM of amoxapine specifically blocks the hydrolysis of d-glucuronide by *S.* Montevideo, thereby reducing the availability of free-DGA for the propagation of *S.* Montevideo without showing toxicity to the bacterial cells. The EC_50_ of amoxapine for *S.* Montevideo was 2.1 µM (Figure 8e). The EC_50_ of amoxapine for *S.* Montevideo in this study was higher when compared with the previously reported EC_50_ of amoxapine for *E. coli* GUS (EC_50_ ranging from 58.5 nM to 119 nM) [31,34]. This is likely due to the differences in the assay conditions. For instance, the concentration of d-glucuronide (1 mM) used in this study was significantly higher when compared with the concentrations used for *E. coli* (125 µM) in the published studies. Moreover, we incubated the assay for an overnight period, whereas, in other studies, the assay incubation time varied from 2 to 6 h [31,34]. Despite these differences, our data clearly shows that amoxapine can efficiently inhibit GUS-mediated hydrolysis of d-glucuronide PNPG by *S.* Montevideo.

### 2.9. Amoxapine Inhibits the Ability of GUS-Positive NTS Serotypes to Hydrolyze d-Glucuronide AA-Gluc in a Serotype-Independent Manner

Here, we tested if amoxapine can also inhibit the ability of other GUS-positive serotypes such as *S.* Schwarzengrund to hydrolyze d-glucuronide AA-Gluc. To accomplish this, *S.* Schwarzengrund was grown for 24 h in M9 containing 1 mM of d-glucuronide AA-Gluc with or without the addition of 10 µM of amoxapine. For comparison, we included GUS-positive control *S.* Montevideo str. 2 and an additional strain of *S.* Montevideo str. 27002. In the presence of amoxapine, the growth of *S.* Schwarzengrund (1.54 log_10_ with Td of 4.45 h) and *S.* Montevideo str. 27002 (1.07 log_10_ with Td of 5.09 h) was significantly (*p* < 0.05) impaired when compared with their counterparts grown in the absence of amoxapine (Figure 9). Similarly, significant impairment in growth was observed for the *S.* Montevideo str. 2 used as control (1.52 log_10_ with Td of 4.53 h). These results show that both serotypes can hydrolyze d-glucuronide AA-Gluc and thereby release free-DGA, which is used as a sole energy source for growth. The addition of amoxapine inhibits GUS-mediated hydrolysis of d-glucuronide AA-Gluc by GUS-positive serotypes, thereby limiting the availability of free-DGA as an energy source for their growth, resulting in significant growth defects.

### 2.10. Combination of Phenelzine and Amoxapine Inhibits the Ability of NTS Serotypes to Derive Energy for Growth from D-Glucuronide and TYR

Considering that TYR and d-glucuronide are both present in the GI tract and can be concurrently utilized by *Salmonella* as energy sources for growth, we tested the growth of *Salmonella* under similar conditions. For this assay, *S.* Montevideo str. 2 (GUS+), *S.* Typhimurium_22711 (GUS-), and *S.* Enteritidis str. 1 (GUS-) strains were grown in medium supplemented with both micronutrients, TYR and AA-Gluc, with or without the combination of phenelzine and amoxapine. As expected, in the absence of inhibitors, all three strains grew up to 5.56 (*S.* Montevideo), 3.87 (*S.* Typhimurium), and 1.54 (*S.* Enteritidis) log_10_ with a doubling time of 4.47, 5.33, and 9.04 h, respectively (Figure 10). When treated with phenelzine alone (30 µM), the growth of *S.* Typhimurium and *S.* Enteritidis was significantly impaired because these serotypes can only utilize TYR as an energy source (Figure 10b,c). In contrast, treatment with phenelzine (30 µM) alone did not negatively impact the growth of *S.* Montevideo because this serotype is also able to hydrolyze d-glucuronide AA-Gluc and, as a result, can utilize the free-DGA as an energy source for growth (Figure 10a). Interestingly, the addition of amoxapine (10 µM) alone significantly reduced the growth of *S.* Montevideo. This was an unexpected result considering that amoxapine is a GUS inhibitor, thus *S.* Montevideo should still be able to efficiently utilize TYR as an energy source for growth in this assay. The underlying mechanism for this phenotype is currently unknown; however, it has been reported that supplementation of TYR with other preferred sources of carbon such as glucose can repress TYR oxidoreductase TynA in *E. coli* [22]. Moreover, *tynA* promoters can be induced when TYR was added in conditions containing a non-preferred carbon source (e.g., glycerol). Thus, it is possible that, in the culture conditions used in this study, AA-Gluc likely serve as preferred carbon source resulting in repression of TYR oxidoreductases leading to impaired growth of *S.* Montevideo. On the other hand, amoxapine does not significantly impair the growth of *S.* Typhimurium (1.09 log_10_ and Td:8.12 h) and *S.* Enteritidis (1.96 log_10_ and Td:10.31 h) since these serotypes do not carry GUS but are still capable of activating the TYR oxidoreductases to metabolize TYR. Finally, the combination of amoxapine and phenelzine completely inhibits the growth of *Salmonella* when TYR and AA-Gluc are used as sources of energy for growth (Figure 10a–c), suggesting that phenelzine efficiently inhibits consumption of TYR and amoxapine reduce the availability of free-DGA utilized by *Salmonella* for propagation in vitro. Overall, the findings of this study should serve as the foundation for the testing combination of phenelzine and amoxapine in vivo, which could potentially be utilized as an anti-nutritional strategy to inhibit the propagation of *Salmonella* in the host.

## 3. Materials and Methods

### 3.1. Bacterial Strains

According to CDC, except *S.* Kentucky, all other *Salmonella* serotypes used in this study (Table 2) are identified as the most clinically significant serotypes associated with food-borne illness in the USA [40,41]. Unless otherwise specified, frozen stocks of these *Salmonella* strains were cultured on Luria–Bertani (LB) agar at 37 °C for 16 h. From each agar plate, a single colony was selected and then inoculated in 5 mL of minimal salt media (M9) containing 20 mM of glucose, followed by incubation at 37 °C for 16 h with shaking at 180 rpm [11,43]. Aliquots (1 mL) from the overnight culture were washed three times with M9 salts. A starting inoculum of ~200 CFU was prepared by serial 10-fold dilution in M9 medium.

### 3.2. Cloning, Expression, and Purification of Recombinant SEN2971, SEN3065, and SEN2426

The open reading frames encoding putative oxidoreductases SEN2971, SEN3065, and SEN2426 were amplified from the genomic DNA extracted from *S.* Enteritidis str. CDC_2010K_0968, using the primers listed in Table 3 and the 2X Platinum SuperFi master mix (Invitrogen, Waltham, MA, USA) following the manufacturer’s instructions (Appendix A). Thermal cycling conditions on an iCycler (Bio-Rad, Hercules, CA, USA) included 1 cycle at 98 °C for 30 s, 30 cycles at 98 °C for 10 s, 58 °C for 10 s, 72 °C for 30 s, and 1 cycle of 72 °C for 5 min. The PCR products were purified with GeneJet PCR purification Kit (ThermoFisher Scientific, Waltham, MA, USA) followed by ligation into pET-100 Directional-TOPO (Invitrogen) according to the manufacturer’s protocol. The recombinant plasmids (pET100::SEN2971, pET100::SEN3065, and pET100::SEN2426) were used to transform TOP10 chemically competent cells (Invitrogen). The transformed TOP10 cells were plated into LB agar containing 100 µg/mL of carbenicillin and incubated overnight at 37 °C. The transformants were screened for the presence of the recombinant plasmid using gene-specific PCR as described above (Table 3). After confirmation, the recombinant plasmids (pET100::SEN2971, pET100::SEN3065, and pET100::SEN2426) were extracted with GeneJET Plasmid Miniprep Kit following the manufacturer’s instructions (ThermoFisher Scientific).

For recombinant protein expression, BL21*DE3 cells (Invitrogen) were transformed with each of the purified recombinant plasmids (pET100::SEN2971, pET100::SEN3065, and pET100::SEN2426). The transformation reaction was transferred to 10 mL of LB broth containing carbenicillin (100 µg/mL) and incubated overnight at 37 °C with shaking (200 rpm). Next, 200 mL of LB broth was inoculated with 4 mL of the overnight culture and then allowed to grow to mid-log phase for approximately 1.5 h at 37 °C with shaking. The recombinant protein expression was induced by the addition of 0.5 mM of IPTG followed by overnight incubation at 20 °C with shaking. The induced culture was centrifuged at 4600 rpm for 15 min and the cell pellets were stored at −80 °C until further use. The cell pellet was resuspended with 8 mL of 1× native binding buffer containing 8 mg of lysozyme (ThermoFisher Scientific) and incubated on ice for 30 min. Subsequently, the cells were lysed by 3 freezing (liquid nitrogen) and thawing (42 °C) cycles and then centrifuged at 3000 g for 15 min. The recombinant proteins were purified from the supernatant using ProBond Purification System (ThermoFisher Scientific) following the manufacturer’s instructions. The purified proteins were eluted in the native elution buffer, evaluated by SDS-PAGE (Appendix A), and stored at 4 °C until further use.

### 3.3. Determination of Oxidoreductase Activity of Recombinant SEN2971, SEN3065, and SEN2426

#### 3.3.1. Radiometric Assay

The oxidoreductase activity of recombinant SEN2971, SEN3065, and SEN2426 was measured by a radiometric assay using [ring-3, 5-3H]-tyramine hydrochloride (American Radiolabeled Chemicals, St. Louis, MO, USA, Catalog # ART 0425). The reaction buffer consisted of the following: 50 mM of monosodium phosphate, pH 7.5, 10 µM of nonradioactive tyramine hydrochloride (Sigma-Aldrich, St. Louis, MO, USA), and 0.051 nM of [ring-3, 5-3H]-tyramine hydrochloride, with or without 100 µM of phenelzine (Sigma-Aldrich, USA, Catalog # CDS015167). The reaction started by adding 100 µl of the purified proteins, and the mixture was kept at room temperature for 1 h. The deaminated products were extracted with 1 mL of toluene (Sigma-Aldrich), shaken vigorously for 5 min, and centrifuged at 5000× *g* for 5 min. The toluene layer (0.9 mL) was then transferred to a counting vial and mixed with 5 mL of the liquid scintillation cocktail Ecoscint-O (National Diagnostics, Atlanta, GA, USA). The oxidoreductase activity of the purified proteins was detected through the rate of ionization events per min or counts per min (CPM) using a Tri-Carb 2900TR liquid scintillation analyzer (PerkinElmer, Seattle, WA, USA). The data were normalized against the CPM obtained from the negative control reaction (without enzyme and radiolabeled TYR). Each protein was tested in three independent experiments and the data was statistically analyzed using two-way ANOVA with Dunnett’s posthoc.

#### 3.3.2. Colorimetric Assay

The oxidoreductase activity of recombinant SEN2971, SEN3065, and SEN2426 was also measured by the colorimetric Amplex™ Red Monoamine Oxidase Assay Kit (ThermoFisher, Waltham, MA, USA) following the manufacturer’s instructions. Briefly, the reaction was conducted in 96-well clear-bottom plates (Evergreen Sci., Vernon, CA, USA, Catalog #222-8050) and consisted of 100 µL of each purified protein and 100 µL of the assay buffer. In this assay, TYR is used as a substrate, and the oxidoreductase activity of the purified proteins is determined quantitatively by detection of the oxidoreductase-dependent formation of H_2_O_2_, which is measured as the change in absorbance at 562 nm using the EL808 plate reader (BioTek, Winooski, VT, USA). Differential inhibition of oxidoreductase activity of each recombinant protein for TYR was achieved by the addition of clorgyline as MAO-A inhibitor or pargyline as MAO-B inhibitor at a final concentration of 100 µM each. The controls included wells with H_2_O_2_ (positive control) or without the purified proteins (negative control). The data were normalized against the OD_562_ obtained from the negative control. Each protein was tested in three independent experiments and the data were analyzed using two-way ANOVA with Dunnett’s posthoc.

### 3.4. Cell-Based Oxidoreductase Inhibition Assays

For this assay, a starting inoculum of ~200 CFU of each *Salmonella* strain (Table 2) was added to the 2 mL of M9 (without NH_4_Cl) containing 5mM of TYR, with or without 30 µM of phenelzine, in 96-well blocks and incubated at 37 °C for 48 h without shaking. For controls, growth in M9 (with NH4Cl) containing 20 mM of glucose, with or without 30 µM of phenelzine, was used to evaluate the cytotoxicity of phenelzine. Each strain was tested in triplicates in 3 independent experiments. The colony-forming units (CFU) were enumerated at 24 h and 48 h, transformed to log_10_, and the data were analyzed by one-way ANOVA with Dunnett’s posthoc.

### 3.5. Cell-Based GUS Activity and Inhibition Assays

GUS activity of *Salmonella* was determined in a cell-based colorimetric assay by quantitative monitoring the release of p-nitrophenol from 4-Nitrophenyl β-d-glucuronide (PNPG, Sigma-Aldrich, Catalog #N1627), a standard in vitro GUS assay substrate. It is expected that only *Salmonella* strains that express GUS can hydrolyze PNPG. To determine the GUS activity, a single colony of *S.* Enteritidis str. 1 (GUS-negative serotype) or *S.* Montevideo str. 2 (GUS-positive serotype) was inoculated in 5 mL of LB broth, followed by incubation at 37 °C for 16 h with shaking at 180 rpm. In addition, 50 µL of the overnight culture was used to initiate a fresh 5 mL culture in LB followed by incubation at 37 °C with shaking at 180 rpm until an OD_600_ of 0.6 was achieved. The cultures were then washed twice with 50 mM of HEPES (pH 7.4) and concentrated by centrifugation to an OD_600_ of 1. The reaction consisted of 40 µL of the bacterial cells (~3.2 × 10^7^), 20 µL of assay buffer (50 mM of HEPES and 1% DMSO), and 40 µL of 2.5 mM of PNPG (final concentration of 1 mM). As a control, 40 µL (a final concentration of 5 nM) of purified GUS encoded by *E. coli* (G8420-25KU, Sigma-Aldrich, USA) was also included in the reaction for comparison. To determine the EC_50_ of amoxapine (Sigma-Aldrich, Catalog #A129), 20 µL of the assay buffer containing amoxapine was added to achieve five different concentrations (1, 5, 10, 50, and 100 µM). Additional controls included wells without PNPG (background, 0% activity) and wells without amoxapine (100% activity). Plates were incubated overnight at 37 °C and the absorbance at 450 nm was measured using an EL808 plate reader (BioTek). The absorbance data were normalized to the background with 0% activity.

The EC_50_ of amoxapine was calculated using nonlinear regression with GraphPad Prism software (GraphPad Software Inc., La Jolla, CA, USA). Cell survivability in the presence of a different concentration of amoxapine was assessed by plating the 10-fold dilutions of the 100 µl of overnight reactions on LB agar followed by incubation for overnight at 37 °C and the CFU were counted as described above. These data were used to select an optimal concentration of amoxapine for follow-up cell-based assays to determine the GUS inhibitory activity of amoxapine against different *Salmonella* serotypes.

For the cell-based GUS-inhibitory assays, frozen stocks of *Salmonella* strains (Table 2) were cultured on Luria–Bertani (LB) agar at 37 °C for 16 h. A single colony of each strain was inoculated in 5 mL of minimal salt media (M9) containing 20 mM of glucose, followed by incubation at 37 °C for 16 h with shaking at 180 rpm [11,43]. Aliquots (1 mL) from the overnight culture were washed three times with M9 salts. A starting inoculum of ~200 CFU was added to the 2 mL of M9 medium containing 1 mM of P-acetamidophenyl β-d-glucuronide sodium salt (AA-Gluc, Santa Cruz Biotechnology, Dallas, TX, USA, Catalog # sc-222105A) with or without 10 µM of amoxapine, followed by incubation at 37 °C for 24 h. In this assay, AA-Gluc was used as a substrate because the hydrolysis of d-glucuronide PNPG in the cell-based colorimetric assay releases p-nitrophenol, a compound that can be toxic for bacterial growth (data not shown). Each *Salmonella* strain was also grown in M9 containing 1 mM of DGA with or without 10 µM of amoxapine to confirm if all the test *Salmonella* strains can utilize DGA as a sole source of energy for growth and to evaluate the cytotoxicity of amoxapine. Each strain was tested in triplicate in three independent experiments. The CFU was enumerated, transformed to log_10_, and the data were analyzed by one-way ANOVA with Dunnett’s posthoc.

## 4. Conclusions

In this study, we show that *Salmonella* scavenges energy for growth from TYR and DGA, in a serotype-independent manner. Three TYR oxidoreductases encoded by the genes SEN2971, SEN3065, and SEN2426 were confirmed to be involved as key enzymes in the first step of TYR oxidation in *Salmonella*. Among the three TYR oxidoreductases, SEN2971 appears to be most efficient, whereas SEN3065 and SEN2426 are likely less efficient and may serve as alternative oxidoreductases. Our data also show that a few *Salmonella* serotypes including *S.* Montevideo and *S.* Schwarzengrund produce β-glucuronidase (GUS), a key enzyme capable of hydrolyzing d-glucuronide into free-DGA. This mechanism enables these serotypes to hydrolyze D-glucuronides, thereby facilitating the scavenging of free-DGA as energy source for propagation. It is noteworthy that phenelzine (MAO inhibitor) and amoxapine (GUS inhibitor) are used as antidepressant drugs. Antidepressant drugs are being increasingly recognized for their potential anti-microbial activity [43,44,45,46]; however, the mechanisms of antimicrobial activity remain poorly understood. Here, we repurposed phenelzine and amoxapine to inhibit the key enzymes committed to the first steps within the TYR and DGA metabolic pathways in *Salmonella*. Phenelzine significantly blocked the activity of the TYR oxidoreductases SEN2971, SEN3065, and SEN2426, leading to the inability of *Salmonella* to utilize TYR as an energy source for propagation. Amoxapine significantly blocked GUS-mediated hydrolysis of d-glucuronide, reducing availability of free-DGA used by *Salmonella* as energy source for propagation. The results also revealed that the combination of phenelzine and amoxapine efficiently inhibits consumption of TYR and DGA simultaneously, thereby inhibiting the growth of *Salmonella*. Given the potential role of TYR and DGA as an important source of energy for *Salmonella* growth in vivo, the data and the novel approaches used in this study will open new avenues to investigate the role of TYR and DGA in *Salmonella* pathogenesis and nutritional virulence in vivo.

## Figures and Tables

**Figure 1 pathogens-10-00469-f001:**
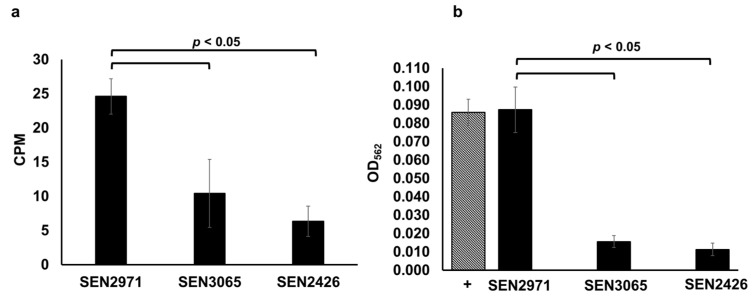
TYR oxidoreductase activity of the recombinant SEN2971, SEN3065 and SEN2426, measured by a radiometric (**a**) and colorimetric assay (**b**). In the radiometric assay, the enzymatic activity of the recombinant proteins is expressed as normalized counts per minutes (CPM). In the colorimetric assay, the MAO-dependent formation of H_2_O_2_ is expressed as normalized OD_562_. The positive control (+) consisted of the addition of 100 µL (10 µM) of H_2_O_2_ (hatched bar).

**Figure 2 pathogens-10-00469-f002:**
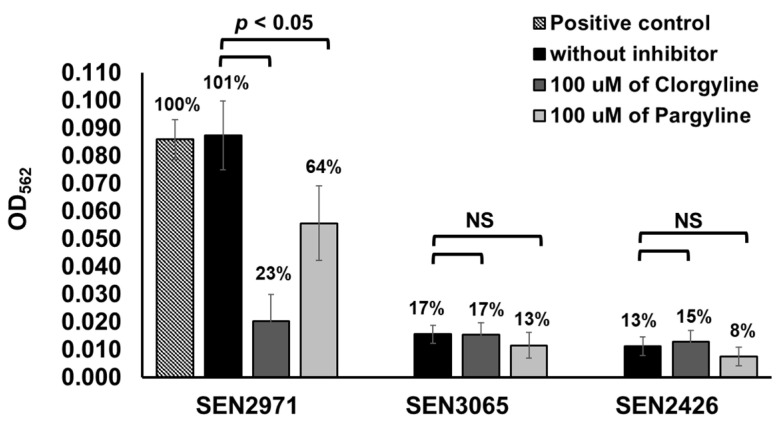
Differential inhibition of TYR oxidoreductase activity of the recombinant SEN2971, SEN3065, and SEN2426 by clorgyline (inhibitor of MAO-A activity) and pargyline (inhibitor of MAO-B activity). The numbers on the top of the bar show percentage of enzymatic activity. NS, non-significant.

**Figure 3 pathogens-10-00469-f003:**
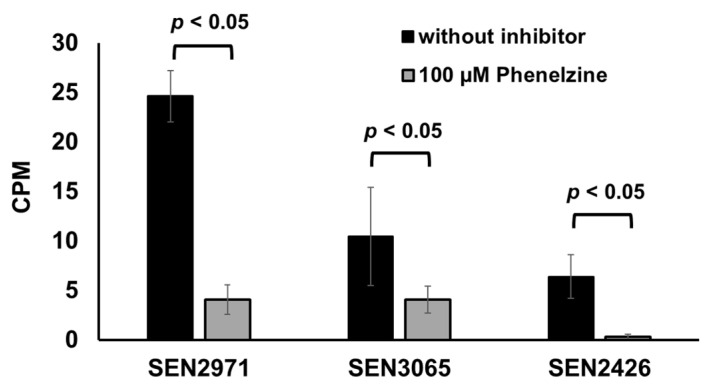
Phenelzine inhibits TYR oxidoreductase activity of recombinant SEN2971, SEN3065, and SEN2426.

**Figure 4 pathogens-10-00469-f004:**
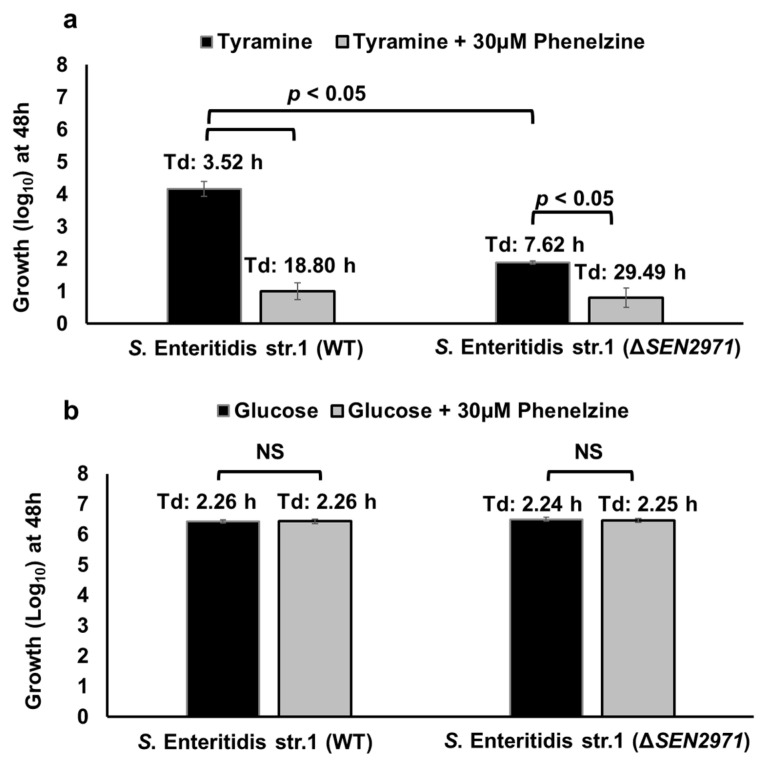
Growth kinetics of *S.* Enteritidis WT and ΔSEN2971 in the presence of TYR (**a**) or glucose (**b**) as a sole energy source, with or without 30 µM of phenelzine. The doubling time (Td) for each strain was calculated using the following formula: t/(3.3 × log(b/B)) where t is the time interval (48 h), b is the CFU at the end of 48 h and B is the CFU at 0 h. Asterisks indicate a significant difference in the Td (*p* < 0.05) when compared with the WT parent. Comparisons with *p*-value > 0.05 are shown as non-significant (NS).

**Figure 5 pathogens-10-00469-f005:**
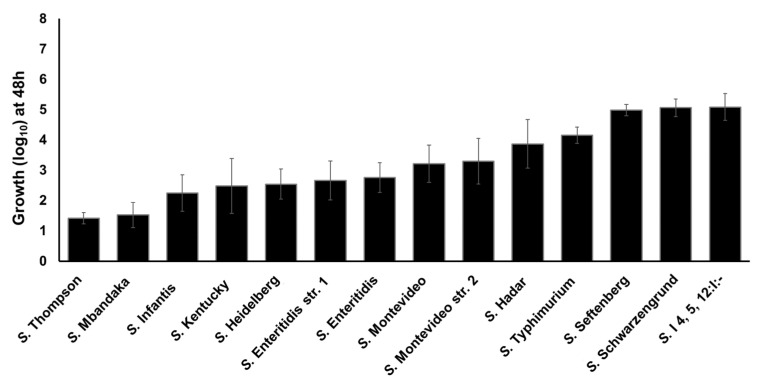
Growth kinetics of the twelve clinically significant NTS serotypes in the presence of TYR as a sole energy source. *S.* Enteritidis str. 1 and *S.* Montevideo str. 2 served as controls.

**Figure 6 pathogens-10-00469-f006:**
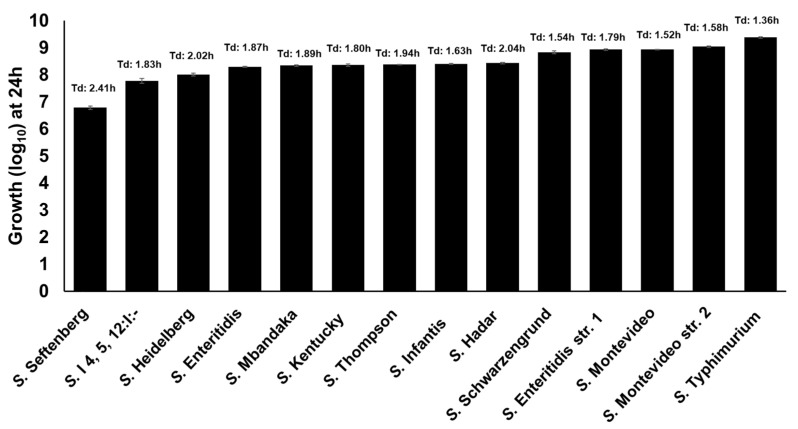
Growth kinetics of twelve clinically significant NTS serotypes in the presence of DGA as a sole energy source. *S.* Enteritidis str. 1 and *S.* Montevideo str. 2 served as controls.

**Figure 7 pathogens-10-00469-f007:**
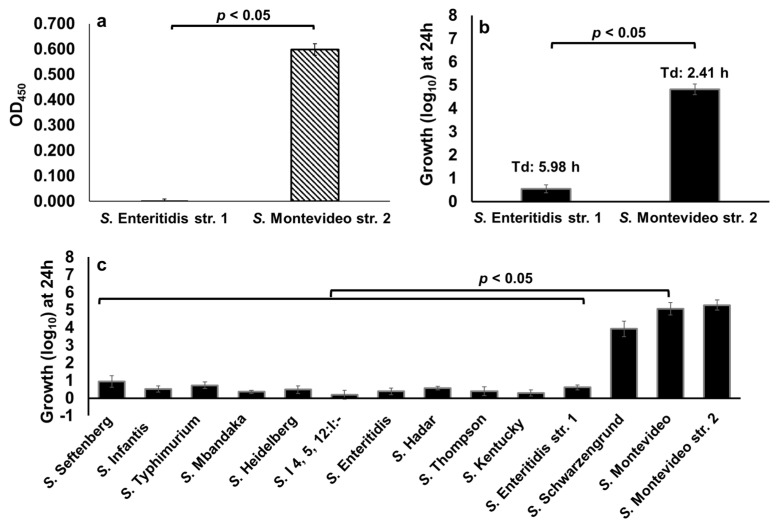
Differential ability of NTS serotypes to hydrolyze d-glucuronide detected through a cell-based colorimetric assay (**a**) and growth kinetics assays (**b**,**c**). Asterisks indicate significant differences (*p* < 0.05). *S.* Enteritidis str. 1 and *S.* Montevideo str. 2 served as controls.

**Figure 8 pathogens-10-00469-f008:**
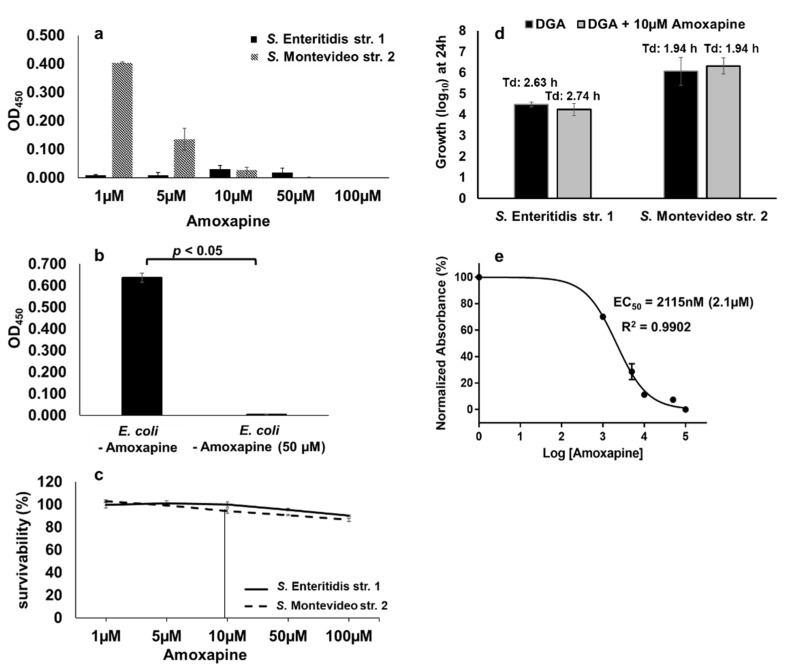
Amoxapine inhibits GUS-mediated hydrolysis of d-glucuronide PNPG by *Salmonella* (**a**–**d**) and *E. coli* (**e**).

**Figure 9 pathogens-10-00469-f009:**
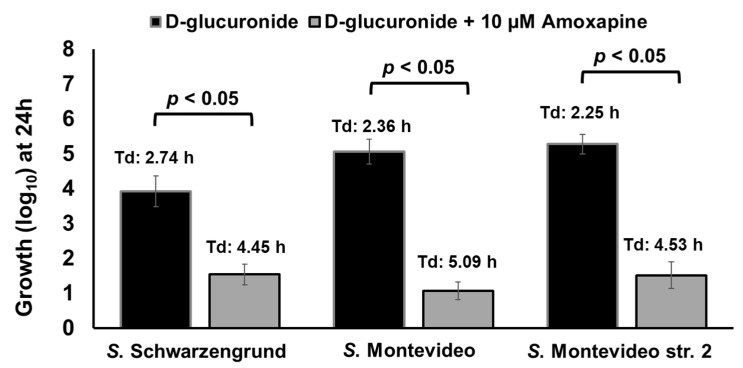
Amoxapine inhibits GUS-mediated hydrolysis of d-glucuronide AA-Gluc by GUS-positive serotypes, thereby limiting the availability of free-DGA as an energy source for growth. Asterisks indicate significant difference (*p* < 0.05). Td, duplication time.

**Figure 10 pathogens-10-00469-f010:**
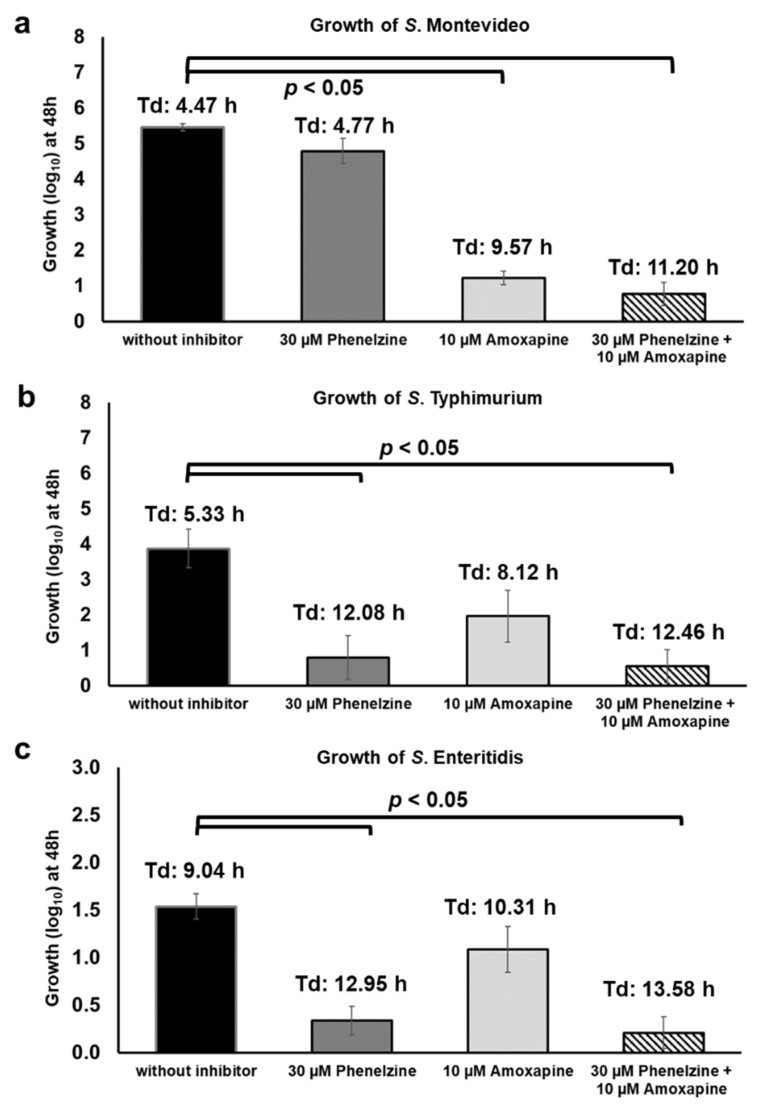
Use of phenelzine, amoxapine, and the combination thereof to inhibit the growth of *S.* Montevideo (**a**), *S.* Typhimurium (**b**), and *S.* Enteritidis (**c**) in the presence of TYR and d-glucuronide AA-Gluc.

**Table 1 pathogens-10-00469-t001:** Doubling time (Td) for the twelve clinically significant NTS serotypes grown in the presence of TYR as a sole energy source, with or without 30 µM of phenelzine. The Td for each strain was calculated as described in Figure 5. Asterisks indicate significant difference (*p* < 0.05). *S.* Enteritidis str. 1 and *S.* Montevideo str. 2 served as controls.

	Doubling Time (h)	
	TYR	TYR + 30 µM of Phenelzine
***S*** **. Thompson**	9.48	13.89 *
***S*** **. Mbandaka**	9.43	13.76 *
***S*** **. Kentucky**	7.89	13.61 *
***S*** **. Infantis**	8.09	13.56 *
***S*** **. Heidelberg**	4.72	14.55 *
***S.*** **Enteritidis str. 1**	7.45	13.61 *
***S*** **. Enteritidis**	7.26	14.12 *
***S*** **. Montevideo**	6.88	13.94 *
***S.*** **Montevideo str. 2**	6.80	13.83 *
***S*** **. Hadar**	6.29	13.27 *
***S*** **. Typhimurium**	5.34	13.89 *
***S*** **. Schwarzengrund**	4.86	13.29 *
***S*** **. Seftenberg**	5.00	14.55 *
***S*** **. I 4, 5, 12:I:-**	4.72	14.55 *

* *p* < 0.05.

**Table 2 pathogens-10-00469-t002:** *Salmonella* strains used in this study.

Strain Number	*Salmonella* Serotype	Strains Used in the Study	Phenotype	AMR Pattern	Reference
1	*S*. Enteritidis str. CDC_2010K_0968	WT and Δ*SEN2971*	GUS-	N/A	Allard et al., 2013; Elder et al., 2018
2	*S*. Montevideo str. USDA_ARS_USMARC-1921	WT	GUS+	N/A	Harhay et al., 2017
3	*S*. Thompson	WT	GUS-	ACSSuTSxtAmcCaz	Shah et al., 2017
4	*S*. Mbandaka	WT	GUS-	ACSSuTAmcCazK	Shah et al., 2017
5	*S*. Kentucky	WT	GUS-	AAmcCCipKNalSSxtT	Shah et al., 2017
6	*S*. Infantis	WT	GUS-	ACGSSuTSxtAmcCaz	Shah et al., 2017
7	*S*. Heidelberg	WT	GUS-	ACSSuTSxtAmcNalCaz	Shah et al., 2017
8	*S*. Enteritidis	WT	GUS-	ACKSSuTSxtAmcCaz	Shah et al., 2017
9	*S*. Hadar	WT	GUS-	ACKSSuTAmc	Shah et al., 2017
10	*S*. Typhimurium	WT	GUS-	ACKSSuTSxtAmcNalCaz	Shah et al., 2017
11	*S*. I4,4,5,12:I:-	WT	GUS-	ACSSuTAmcCaz	Shah et al., 2017
12	*S*. Seftenberg	WT	GUS-	AKSSuTAmcCaz	Shah et al., 2017
13	*S*. Schwarzengrund	WT	GUS+	ACKSSuTSxt	Shah et al., 2017
14	*S*. Montevideo	WT	GUS+	ASSuTSxt	Shah et al., 2017

**Table 3 pathogens-10-00469-t003:** Primers.

Gene	Primers	Sequence 5′ to 3′	Product Size (bp)
*SEN2971*	RA01clonSEN2971Fw	CACCATGAATACAAAAATCGAT	1302
	RA01clonSEN2971Rv	TTAATTCCGGCCTTTCCAG	
*SEN3065*	RA04clonSEN3065Fw	CACCATGATACGTTTCGCTGTA	999
	RA04clonSEN3065Rv	TTACGCAGTAAGGGGATGA	
*SEN2426*	RA03clonSEN2426Fw	CACCATGGGTAAACTCACGGGC	792
	RA03clonSEN2426Rv	TCAGACGCCTACGCTTACG	

## Data Availability

Data are contained within the article or Appendix A.

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
