# Peer review of "Phenelzine and Amoxapine Inhibit Tyramine and d-Glucuronic Acid Catabolism in Clinically Significant Salmonella in A Serotype-Independent Manner"

_pathogens, 2021, doi:10.3390/pathogens10040469_

Round 1

Reviewer 1 Report

The manuscript by Burin and Shah describes the use of two antidepressants as potential inhibitor of Salmonella strains that are clinically relevant.

I have some comments and suggestions for the authors:

  1. this is not the first time that antidepressants are suggested to be used as antimicrobial, some references to these other manuscripts are missing
  2. how does the active concentration of the two antidepressants used as inhibitors of metabolims compared to the concentration that these drugs are found the human body when used as antidepressants?
  3. the authors purify different proteins, but do not show any SDS-Page gel to show how good the purification was, how pure the protein and do not mentioned how much proteins are used in the assay: 100ul is not enough, if it is not mentioned the protein concentration, which of course should always be the same if comparisons are to be made. This applied to radiometric and colorimetric assays.
  4. the manuscript would benefit from a section "Conclusions"
  5. How were the working concentrations of the two antidepressants established? Was a MIC done?

Minor:

  1. names of microbes is often wrong, for example S. Enteridis, should be written S.enteritidis, and E.coli is E.coli, please check the whole manuscript and the figures
  2. Fig.S1 should be in a seprate file: a supplementary one
  3. seel lanes 409 and 410 for typing errors: Materials and Methods is inserted in the worng spot
  4. p of p-value is "p" not "P"

Author Response

Comments and Suggestions for Authors

The manuscript by Burin and Shah describes the use of two antidepressants as potential inhibitor of Salmonella strains that are clinically relevant.

I have some comments and suggestions for the authors:

  1. this is not the first time that antidepressants are suggested to be used as antimicrobial, some references to these other manuscripts are missing.

Yes, antidepressant drugs have been suggested as having potential antimicrobial activity. Some studies have shown antimicrobial activities in a very high concentrations without explaining the underlying mechanisms. In this study, we show that phenelzine and amoxapine do not actually have direct antimicrobial activity at the concentrations used. However, the effects are mainly due to the MAO and GUS inhibition in media supplemented with the essential nutrients. Thus, our study shows mechanism of the anti-Salmonella activity, however we have avoided overemphasizing/generalizing the antimicrobial activity of these compounds. To highlight the potential use of antidepressants as anti-microbials, we added few sentences in the concluding remarks where we also cite a few references as recommended by reviewer. We included additional citation that was missing in line 112 (Ahmad et al. 2012). We hope that the review will find these additions as satisfactory.

  1. how does the active concentration of the two antidepressants used as inhibitors of metabolims compared to the concentration that these drugs are found the human body when used as antidepressants?

The effective dosage of amoxapine as antidepressant in humans considering an average of human body weight of 70 kg, is 2.85 mg/kg. Studies conducted in mice and that tested the effect of amoxapine in the inhibition of microbial GUS, used a dosage of amoxapine of 2.5 mg/kg twice daily. In our study, we used 10 µM of amoxapine in vitro, which corresponds to 3 mg/kg, a concentration that is very close to what is used in humans and mice studies.  

To obtain a sufficient response as antidepressant in humans, phenelzine should be used with a dosage up to 1.28 mg/kg, also considering an average of human body weight of 70 kg. In our study, we used 30 µM of phenelzine in vitro, which corresponds to 7 mg/kg.

References:

  1. Amoxapine (Amoxapine Tablets): uses, dosage, side effects and drug interactions. https://www.rxlist.com/amoxapine-drug.htm#warnings. Last reviewed on: 12/29/2017.

  1. Kong, R.; Liu, T.; Zhu, X.; Ahmad, S.; Williams, A.L.; Phan, A.T.; Zhao, H.; Scott, J.E.; Yeh, L.A.; Wong, S.T.C. Old drug new use - Amoxapine and its metabolites as potent bacterial β-glucuronidase inhibitors for alleviating cancer drug toxicity. Clin. Cancer Res. 2014, 20, 3521–3530, doi:10.1158/1078-0432.CCR-14-0395.

  1. Nardil (Phenelzine): uses, dosage, side effects and drug interactions. https://www.rxlist.com/nardil-drug.htm#description. Last reviewed on: 10/19/2018.

  1. The authors purify different proteins, but do not show any SDS-Page gel to show how good the purification was, how pure the protein and do not mentioned how much proteins are used in the assay: 100ul is not enough, if it is not mentioned the protein concentration, which of course should always be the same if comparisons are to be made. This applied to radiometric and colorimetric assays.

The recombinant proteins were obtained from three INDEPENDENT experiments conducted on separate days and prepped following the same protocol describe for ProBond Purification System. Subsequently, the concentrations of each recombinant protein were estimated based on the intensity of the bands visualized in the SDS-PAGE gel (a representative gel picture is included as supplementary figure 1). The purity of protein was more than 90%. Considering that all the proteins showed the similar band intensity in the three independent experiments, we used the same volume (100 µl) that is required to run the colorimetric and radiometric assays.

  1. the manuscript would benefit from a section "Conclusions"

      A conclusion section was added as recommended.

  1. How were the working concentrations of the two antidepressants established? Was a MIC done?

The concentration of the phenelzine was established by testing varying concentrations of phenelzine in a growth/survival assay of Salmonella (see supplementary figure included here). The concentrations between 30 to 100 uM did not result in any toxic effects on Salmonella (see the figure included here). The lowest concentration that inhibited MAO activity was 30uM. Thus, we decided to use 30uM. Similarly, the concentration of amoxapine was established by testing varying concentrations for the ability to inhibit GUS without any off target/cytotoxic effects on Salmonella cells (See Fig. 8 in the manuscript).

Minor:

  1. names of microbes is often wrong, for example S. Enteridis, should be written S.enteritidis, and E.coli is E.coli, please check the whole manuscript and the figures

The E. coli name is corrected as recommended. The new nomenclature of Salmonella states that all serovars be cited with italicized genus followed by non-italicized serovar name. This has been clarified in line 98 at the first occurrence of Salmonella serovar.

  1. Fig.S1 should be in a seprate file: a supplementary one

Corrected. We have moved this figure and renamed this as Figure S2

  1. see lanes 409 and 410 for typing errors: Materials and Methods is inserted in the wrong spot.

Corrected.

  1. p of p-value is "p" not "P".

Corrected in text and figures.

Reviewer 2 Report

I think the topic is of interest, but the article should be improved.
First of all, the text should make clear what is the reason for the work, the working methods should be specified more clearly, those sections that are reviews of previous works, as well as what is the reason for doing the work. Perhaps too much information is offered that is difficult for the reader to assimilate, since it is somewhat disordered.
It would be of great interest to clarify what would be the future clinical and therapeutic implications of the study, at what point they are and what would be the future.
Probably with a new structuring of the article, it would be easily publishable. 

Author Response

Comments and Suggestions for Authors

I think the topic is of interest, but the article should be improved.
First of all, the text should make clear what is the reason for the work, the working methods should be specified more clearly, those sections that are reviews of previous works, as well as what is the reason for doing the work. Perhaps too much information is offered that is difficult for the reader to assimilate, since it is somewhat disordered.

The aims of this work were clarified in the introduction section in the original version. Please see lines 70 to 79 and lines 112-116. The review of previous work is important in the introduction section to orient researchers with background. We have made minor adjustment wherever applicable.

It would be of great interest to clarify what would be the future clinical and therapeutic implications of the study, at what point they are and what would be the future. Probably with a new structuring of the article, it would be easily publishable. 

The therapeutic and clinical implications of this work will require further investigations; therefore, we have avoided making bold inferences. However, we highlighted this aspect and the future work in the introduction and conclusion section which has been included within the revised structure (see lines 122-125; 486-489; 667-669).

Reviewer 3 Report

This manuscript describes the discovery and characterization of MAO inhibitor phenelzine and GUS inhibitor amoxapine that significantly inhibits the growth of Salmonella by inhibiting TYR-oxidoreductases and GUS-mediated hydrolysis of D-glucuronides, respectively. Minor questions are listed below.

  1. The format of D-glucuronides should be consistent. For example, line 18 is “d-glucuronides” and line 22 is “D-glucuronides”
  2. For Figure S1, the author tested the present of SEN2971, SEN3065, SEN2426 by agarose gel electrophoresis, it would be better to explain the weaker band of S. Seftenberg for SEN2971.
  3. The description of Figure 10 need to be revised.
  4. There are a few typos need to be fixed in the manuscript. For example, Figure 1, H2O2->H2O2, line 231, NH4Cl->NH4Cl

Author Response

Reviewer-3

Comments and Suggestions for Authors

This manuscript describes the discovery and characterization of MAO inhibitor phenelzine and GUS inhibitor amoxapine that significantly inhibits the growth of Salmonella by inhibiting TYR-oxidoreductases and GUS-mediated hydrolysis of D-glucuronides, respectively. Minor questions are listed below.

  1. The format of D-glucuronides should be consistent. For example, line 18 is “d-glucuronides” and line 22 is “D-glucuronides”

Corrected in text and figures.

  1. For Figure S1, the author tested the present of SEN2971, SEN3065, SEN2426 by agarose gel electrophoresis, it would be better to explain the weaker band of S. Seftenberg for SEN2971.

Since the negative control worked well, we did not see the need to re-run the PCR for SEN2971 in S. Seftenberg. It is possible that the faint band observed for this strain occurred because of a lower amount of gDNA used in the PCR or lower amount of PCR reaction added in the well. Irrespective of the reason, the presence of faint band is sufficient to confirm the presence of gene.  

  1. The description of Figure 10 needs to be revised.

The legend was revised as recommended.

  1. There are a few typos need to be fixed in the manuscript. For example, Figure 1, H2O2->H2O2, line 231, NH4Cl->NH4Cl

Corrected in text and legends or figures.

Round 2

Reviewer 1 Report

The manuscript layout improved with this version, but I still have a majour concern with the SDS Page and the protein concentration. The image s1 shows that the proteins are indeed relatively pure BUT:

SEN2971 is the least pure and still the most effective

the bands have different intensity, between SEN2971 and 2426 probably even 10 fold (!!!) and the activity cannot be compared as such, but the protein concentration for each extraction should be used as a reference, therefore expressing everything per mg or ug of protein.

In this way the differences already seen between the activity of the different  proteins will be even more significant.

I think this is a majour point considering how much of the work presented is based on these purified proteins.

Author Response

It is important to note that each independent replicate within each recombinant protein assay conducted in this study required large culture volumes (200 ml) which yielded very small quantities of purified proteins (often recovered from 3rd or 4th elution). Given this limitation, we decided to use the intensity of gel bands which were roughly similar, although this may not be appropriately reflected in scanned pictures of individual gels. It is important to note that for each assay, we used recombinant protein produced separately and each assay was conducted at least three times with appropriate controls, and this consistently resulted in similar outcomes (Fig. 1, 2, and 3). This level of replication and consistency in the outcomes shows that the data is reliable. What’s more important to recognize is that irrespective of protein concentrations used, the primary outcome of this research is that our data clearly shows that all three proteins are indeed TYR oxidoreductases and that phenelzine inhibits all three TYR oxidoreductases. It is possible that the potential difference in protein concentrations may have had confounding effects on our comparison of the efficiency of three TYR oxidoreductases, so while we respectfully agree with reviewers' concern, we believe that the limitation raised by the reviewer is valid, but a minor point. We have appropriately addressed this limitation in this revised version (see lines 148-152) which we hope that the reviewer will find satisfactory.  

Reviewer 2 Report

The work submitted for further review shows improvement.  The indications of the reviewers, who recommended modifying some sections of the work, have been followed. 

Author Response

Thank you for your valuable input and recommendation that the revisions made by us improved the quality of this manuscript to the acceptance level for publication in Pathogens. 

Round 3

Reviewer 1 Report

No more comments